environmental chemistry/inorganic chemistry/nanotechnology

nanoscale zero-valent iron, expanded graphite, Cr (VI), physical adsorption, chemical reduction

**Authors for correspondence:**
Yangshuai Qiu
e-mail: qiuyangshuai@whut.edu.cn
Xuan Jiao
e-mail: x.jiao@seu.edu.cn

# Nanoscale zero-valent iron loaded vermiform expanded graphite for the removal of Cr (VI) from aqueous solution

Xinwei Cai[1,2], Yangshuai Qiu[1,2], Yanhong Zhou[1,2] and Xuan Jiao[3]

[1]School of Resources and Environmental Engineering, and [2]Hubei Province Key Laboratory for Processing of Mineral Resources and Environment, Wuhan University of Technology, Luoshi Road 122, Wuhan, Hubei 430070, People's Republic of China
[3]School of Materials Science and Engineering, Southeast University, Southeast Road 2nd, Nanjing, Jiangsu 211189, People's Republic of China

XJ, 0000-0002-8464-4238

Cr (VI) is indispensable in industrial manufacturing, and its extensive use leads to severe heavy-metal pollution in the water environment around people, posing a great danger to physical health and living environment of multitudinous organisms. Expanded graphite (EG) is considered as a typical material for adsorption, while nanoscale zero-valent iron (nZVI) can be applied to degrade and sedimentate various organic or inorganic pollutants. In this study, a simultaneous collaboration of EG and nZVI is carried out, with the investigation on the influence of different test conditions for adsorption performances. These findings demonstrate that nZVI@EG manifests favourable adsorptive performance on the removal of hexavalent chromium efficiently. nZVI, acting as an electron donor, is supposed to reduce Cr (VI) to Cr (III), turning itself into iron oxide or hydroxide. The whole process is an exothermic reaction, accompanying chemical reduction and physical adsorption. And Cr (III) is fastened on the appearance by deposition of chromium hydroxide or ferrochromium complex precipitation, which greatly reduces the total chromium content in the aqueous solution. Herein, as a new composite adsorbent, nZVI@EG shows promising prospects of practical applications in water contamination and environmental remediation.

This article has been edited by the Royal Society of Chemistry, including the commissioning, peer review process and editorial aspects up to the point of acceptance.

# 1. Introduction

Heavy metals, referring to the metallic elements with a specific gravity greater than 4 or 5 g cm$^{-3}$, are extreme toxic to organisms [1–3]. Except atmospheric and soil ecosystems, water pollution

resulting from heavy metals is a severe challenge faced in environmental protection [4,5]. After entering into water, heavy metals pollutants mainly migrate and transform through a series of physical and chemical actions, including precipitation, complexation and colloid formation, finally existing in one or several stable forms. Owing to the stability and persistence in the food chain, they are hard to destroy and degrade in the process of metabolism of living things [6–8]. And the rapid development of industry is able to impute to the excessive release of heavy metal elements in the areas of metallurgy, battery production, electroplate and mining, bringing a great harm to soil system, and ultimately, harming human beings [9–11].

Among numerous kinds of heavy metal elements, Cr (VI) and its compounds are listed as one of the environmental priority control pollutants. Hexavalent chromium can be absorbed and retained in the body with waste, and a small amount of exposure will cause nasal mucosal discomfort, ulcers or nasal septum perforation, while long-term exposure to Cr (VI) is prone to 'chromium lung cancer', intestinal diseases and anaemia. Therefore, it reveals a vital significance and urgency to reduce the contents of Cr (VI) in aquatic environment to an approved level for life health and safety. In recent researches, several advances spring up on the treatments to remove Cr (VI), such as physico-chemical adsorption, ion exchange under electric field, sol–gel and membrane separation [12–15]. For large-scale application and production cost reduction, adsorption technique is regarded as the most effective method [16–18].

Expanded graphite (EG) is a typical derivative of graphite, obtained from graphite intercalation compounds after the process of extending along the crystallographic c-axis at a high temperature [19,20]. Like plentiful graphitic products, EG presents remarkable chemical durability and nice hydrophobicity. Ascribed to the specific structure, EG equips with abundant networks and particular wormlike forms, promoting EG to be a promising alternative for adsorbent materials [21–23]. Furthermore, nanoscale zero-valent iron (nZVI), first synthesized in the 1990s, is considered as a promising alternative for adsorptive applications in the *in situ* remediation of contaminative edatope and water circumstance treatment, based on the high reactivity and large specific surface area [24–26]. Formerly, various modification processes have been conducted on EG to heighten the processing ability of adsorption. Jin *et al.* confirmed that $MnO_2$-loaded EG displayed high removal efficiency on Cr(VI) [27], and in our previous study, sulfonated EG was prepared via mechanochemical treatment to adsorb Pb (II) in aqueous solution [28]. In addition, to achieve a noteworthy enhancement on the stability and mobility of nZVI, a variety of carbonaceous materials, like carboxymethyl cellulose [29], carbon nanotube [30] and tubular nitride carbon [31] were applied as the substrate for better attachment. However, the simultaneous collaboration of EG and nZVI for the application on the adsorption removal for heavy metals was rarely reported.

The objective of our research was to use chemical-deposited modification to prepare nZVI-loaded EG, served to be a resultful adsorption material to removal Cr (VI) in aqueous solution. The systematic experimental tests were carried out at various conditions, investigating removal performance behaviours on the basis of adsorptive isotherms, kinetics and thermo-dynamics experimental testing and theoretical fitting. Finally, the explanations on mechanism of Cr (VI) treatment were clarified relying on morphological characterization and chemical analysis.

# 2. Material and methods

## 2.1. Materials and chemicals

The natural graphite with fine flakes, keeping an average size of 0.18 mm, was selected after flotation from a graphite mine in Henan, China. Hydrochloric acid (HCl), sodium hydroxide (NaOH), Perchloric acid ($HClO_4$), acetic acid ($CH_3COOH$), sodium borohydride ($NaBH_4$), ferrous sulfate ($FeSO_4 \cdot 7H_2O$), ethanol, polyethylene glycol (PEG), MF dispersant and potassium dichromate ($K_2Cr_2O_7$) were purchased from Sinopharm Chemical Reagent Co., Ltd (China). All regents were of analytical grade and used without further purification.

## 2.2. Preparation of expanded graphite

The natural graphite powders were added into the mixed acids composed of $HClO_4$ and $CH_3COOH$ (3 : 1, w/w), stirring at the temperature of 50°C for 1 h with the mass ratio of 1 : 5. Afterwards, the as-prepared graphite treated with oxidation and intercalation reactions was filtrated, washing with deionized water to pH = 7, drying at 80°C in a vacuum oven. And expanded graphite (EG) was obtained by heating the acid-treated graphite to 800°C in a muffle furnace for several seconds with expanded volume of 280 ml $g^{-1}$.

## 2.3. Preparation of nanoscale zero-valent iron loaded expanded graphite

The liquid-phase reduction method was used for the fabrication of nanoscale zero-valent iron loaded expanded graphite (nZVI@EG). $FeSO_4 \cdot 7H_2O$ (1.0 g) was added into the ethanol solution (40%), then 0.5 g EG was immersed into the ferrous solution for 24 h. Subsequently, 0.3 g PEG and MF dispersant was added into the system with mechanical stirring for 30 min under nitrogen atmosphere. And nZVI@EG was acquire after adding a little $NaBH_4$ solution dropwise for 30 min reaction, dried at 80°C.

## 2.4. Characterization

A JSM-IT300 scanning electron microscope (SEM) was used to examine the surface morphology (JEOL, Tokyo, Japan). X-ray diffraction (XRD) patterns were determined by a D8 Advance model X-ray powder diffractometer (Bruker Corporation, Stuttgart, Germany) with Cu Kα radiation. The analysis of surface functional group was observed via Fourier transform infrared (FT-IR) spectra, recorded with a Nicolet IS-10 infrared spectrophotometer (Nicolet Corporation, Madison, USA) ranged from 400 to 4000 cm$^{-1}$. The VG Multilab 2000 spectrometer (Thermo Electron Corporation, Waltham, MA, USA) was used for X-ray photoelectron spectroscopy (XPS) testing. And the specific surface areas were calculated via Brunauer–Emmett–Teller (BET) method, using an ASAP 2020 M $N_2$ adsorption–desorption apparatus (Micromeritics Instrument Corporation, Atlanta, USA).

## 2.5. Adsorption experiments

A series of adsorptive processes were conducted in 150 ml conical flasks, with 50 ml aqueous solution containing Cr (VI) in each bottle. $K_2Cr_2O_7$ (0.2829 g) was dissolved into 1 l deionized water for the preparation of standard Cr (VI) solution. And pH values of the solution were regulated by the addition of HCl and NaOH solutions (1 mol l$^{-1}$). The adsorptive experiments were carried out in a controlled shaker at 150 r.p.m., under an initial metal concentration of 10 mg l$^{-1}$ Cr (VI). Afterwards, nZVI@EG was separated via filtration after complete adsorption, and an ultraviolet spectrophotometer (540 nm) was used to evaluate the concentration of remaining Cr (VI). Removal efficiency was calculated based on the equation as below.

$$\eta\,(\%) = \frac{C_0 - C_t}{C_0} \times 100\%, \tag{2.1}$$

where $C_0$ and $C_t$ are initial and equilibrium concentrations of Cr (VI) in the solution, respectively.

# 3. Results

## 3.1. Preparation and characterization of nanoscale zero-valent iron loaded expanded graphite

Owing to the weak requirement on manufacturing facility of liquid-phase reduction, the reaction conditions can be better controlled, which is the most common approach for the synthesis on nZVI in current research. As shown in figure 1a, the abundant porous structure and favourable adsorption capacity made it possible for ferrous ions to be adsorbed onto the surface and pores of EG during the soaking process. And the reduction of attached ferrous ions was proceeded to generate nZVI with the presence of sodium borohydride, ultimately obtaining expanded graphite-supported nZVI (nZVI@EG) after several cycles of deionized water washing. The surface morphology of EG is presented clearly in figure 1b,c, displaying obvious vermiform appearance and fluffy chain structure. Oxidized and intercalated by mixed acids, the distance between layers of EG increased to a large extent after calcining at high temperature. Forming a specific pore structure, the surface of EG was smooth without other impurities existing. On the contrary, the rough structure occurred on the surface of EG attributed to the chemical deposition of spherical and granular nZVI nanoparticles upon the external and pore structure, with uniform distribution and no conspicuous agglomerations (figure 1d).

The phase composition of pristine and nZVI-loaded EG was surveyed by XRD in figure 2a. For EG, two characteristic diffraction peaks of natural graphite were retained, an intense peak with $2\theta = 25.8°$ corresponding to the diffraction of 002 planes and a weaker peak with $2\theta = 53.7°$ corresponding to the diffraction of 004 planes. Compared with original samples, a particularly sharp characteristic diffraction peak of α-Fe appeared at approximately 45° on nZVI@EG pattern, suggesting that nZVI

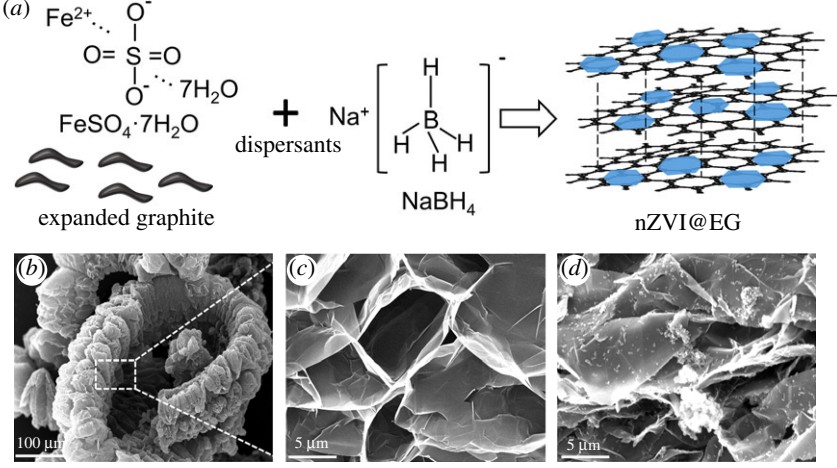

**Figure 1.** A schematic of nZVI@EG and surface morphology. (*a*) Schematic illustration on the synthesis process of nZVI@EG through chemical deposition. (*b,c*) SEM images of vermiform (EG) with different magnification. (*d*) SEM image of nZVI@EG.

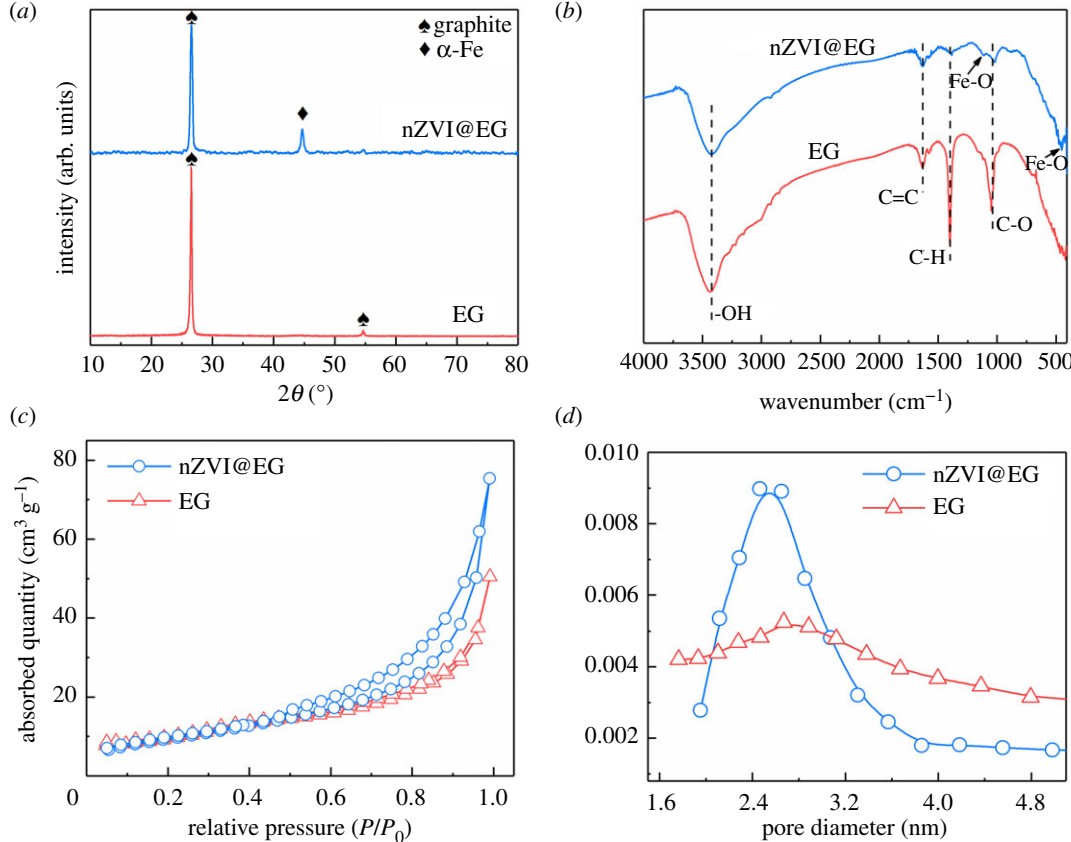

**Figure 2.** Characterization of EG and nZVI@EG. (*a*) XRD patterns of EG and nZVI@EG. (*b*) FT-IR spectra of EG and nZVI@EG. (*c*) Nitrogen adsorption and desorption isotherm of EG and nZVI@EG at 77 K. (*d*) The pore size distributions of EG and nZVI@EG according to BJH (Barrett–Joiner–Halenda) method with desorption summary.

nanoparticles were successfully loaded onto the appearance of EG in a crystal form. The main functional groups of EG before and after modification were determined via FT-IR spectra in figure 2*b*. For the two specimens, the distinct adsorption peaks at 3425 and 1631 cm$^{-1}$ were found, which were connected with the stretching vibrations of –OH from crystal water and C=C stretching mode, respectively [32,33]. And the bands approached to 1394 cm$^{-1}$ could accuse to the asymmetric vibration of C–H in –CH$_3$ groups [34], while the adsorptive intensity of nZVI@EG decreased significantly due to the massive nanoparticles covered the whole surface. In addition, new peaks turned up at the positions of 476 and 1022 cm$^{-1}$, identifying with the stretching vibrations of Fe–O, which implied the effective deposition

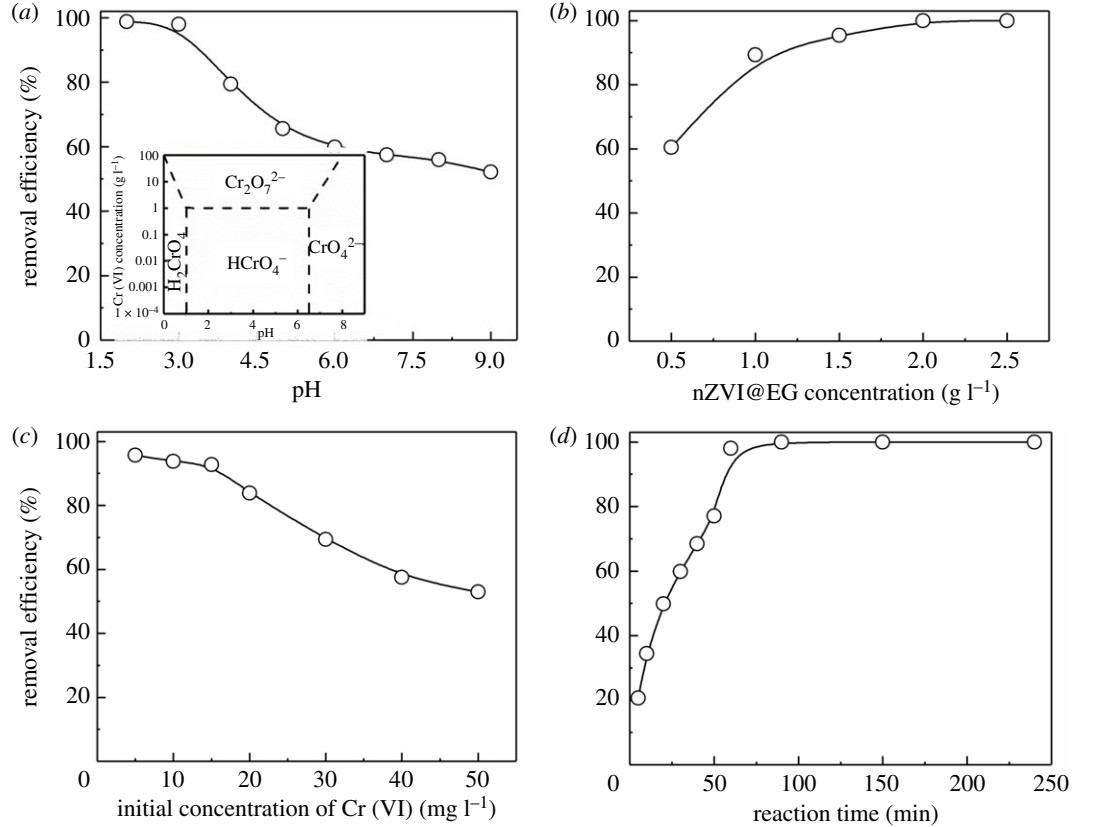

**Figure 3.** Adsorption tests on the removal of Cr (VI). (a–d) The effects of environmental pH, adsorbent dosage, initial concentration and contact time on the removal efficiency of Cr (VI) in aqueous solution, respectively, inserting the form distribution of Cr (VI) at different pH.

of nZVI nanoparticles. For adsorbent materials, the changes on specific surface areas and pore size distributions were critical factors to the modification project. The $N_2$ adsorption and desorption isotherms of EG and nZVI@EG were observed in figure 2c. And the specimens both followed a type IV isotherm on the basis of Brunauer–Deming–Deming–Teller (BDDT) classification [35]. The evident non-overlap appeared at a relative pressure ranged from 0.6 to 1.0, due to the typical capillary condensation phenomenon on nitrogen during the processes of both adsorption and desorption to the formation of hysteresis loops. Based on the results, the specific area of original EG was $32.11 \, \text{m}^2 \, \text{g}^{-1}$, while an improvement was achieved on nZVI@EG, with a specific area of $45.32 \, \text{m}^2 \, \text{g}^{-1}$, indicating more adsorption sites to enhance adsorptive capacity. From figure 2d, EG contained a variety of pore sizes and a broad distribution. Inversely, the pore sizes of nZVI@EG were concentrated in small range of 2–3 nm, resulted from the nanoparticles filling in large size pores.

## 3.2. Adsorption study of Cr (VI) on nanoscale zero-valent iron loaded expanded graphite

Heavy metal ions acting as adsorbates, the surface adsorption properties were greatly affected by the pH of solution environments [36]. Cr (VI) ions could generate precipitation and attach to the surface of the absorbent material, preventing the reduction on nZVI nanoparticle surface at too high pH in the reaction system. From figure 3a, the removal efficiency of Cr (VI) cut down successively as the pH in the solution increasing. And the reaction potential could be calculated based on the following equations:

$$Fe^0 + 2H^+ \rightarrow Fe^{2+} + H_2 \ (pH \le 7), \tag{3.1}$$

$$E_h = E_0 - \frac{0.0592}{n} \lg\left(\frac{[Fe^{2+}]}{[H^+]^2}\right), \tag{3.2}$$

$$Fe^0 + 2H_2O \rightarrow Fe^{2+} + H_2 + 2OH^- \ (pH > 7) \tag{3.3}$$

and

$$E_h = E_0 - \frac{0.0592}{n} \lg\left([Fe^{2+}][OH^-]^2\right), \tag{3.4}$$

where $E_0$ is the standard reduction potential, and $n$ is the number of reacting electrons. Under acidic conditions, the high concentration of hydrogen ions enhanced the reduction potential, leading to a greater

activity of nZVI on the surface. Meanwhile, the main form on the presence of Cr (VI) was $HCrO_4^-$, improving the adsorptive capacity by the electrostatic attraction. Figure 3b investigated the effect on different amounts of adsorbent to remove Cr (VI). More additive amount was supposed to result in greater removal efficiency; however, increasing the cost of production simultaneously. From the results, the removal efficiency raised with the increment on the concentration of nZVI@EG, and adequate adsorbents were able to ensure high adsorptive performance as far as possible. The remarkable adsorption effect of nearly 100% removal efficiency was acquired at the condition of 2 g l$^{-1}$ nZVI@EG. The oxidation reduction between nZVI and Cr (VI) was crucial during the whole treatment process, assisted by physical adsorption on EG. Excessive nZVI@EG amount could cause unsaturated adsorption of absorbents in the solution, reducing the service efficiency on adsorptive sites with a waste of resources in practical application. And the influence on initial concentration of adsorbed ions was an essential factor to explore the adsorption reaction. Apparently to be found that the removal efficiency decreased continuously, with the increase on initial Cr (VI) concentration. Shown in figure 3c, removal efficiency was below 55% while Cr (VI) concentration reaching 50 mg l$^{-1}$. More active sites were exposed on the surface of nZVI@EG in the primary stage of adsorption at the condition of low Cr (VI) concentration. And the diffusion driving force of hexavalent chromium ions in the aqueous solution was enhanced with the increasing initial Cr (VI) concentration, resulting in conspicuous degeneration in removal performance. Thus, for the wastewater with high concentration of heavy metal ions, appropriate improvement on the usage amount was pivotal. For a cost-effective adsorption system, the reaction time of nZVI@EG contacted with Cr (VI) in the solution was a vital parameter for assessment. Figure 3d exhibits that the removal efficiency increased rapidly in the initial phase, with slight variation tendency subsequently and reaching final equilibrium condition. The removal efficiency achieved almost 80% in the first 50 min, the prolonging adsorption time, the increase rate slowed down until reaching saturation capacity, on account of the reduction on the available adsorptive sites and consumption of nZVI nanoparticles.

## 3.3. Adsorption models and thermodynamics

In order to accurately elaborate the relevancy on adsorbate concentration and adsorption amount of absorbent in equilibrium stage, the Langmuir [37–39] (equation (3.4)) and Freundlich [40,41] (equation (3.5)) isotherms were applied for theoretical calculation.

$$\frac{1}{q_e} = \frac{1}{K_L Q_m C_e} + \frac{1}{Q_m} \tag{3.5}$$

and

$$\ln q_e = \ln (K_F) + \frac{1}{n_F} \ln (C_e), \tag{3.6}$$

where $C_e$ represents equilibrium Cr (VI) concentration remained in the solution (mg l$^{-1}$), $q_e$ represents Cr (VI) amount adsorbed by nZVI@EG (mg g$^{-1}$), $Q_m$ is the maximum amount by per unit nZVI@EG (mg g$^{-1}$), $K_L$ expresses the adsorptive enthalpy (l mol$^{-1}$). $K_F$ and $n_F$ are Freundlich constants.

Shown in figure 4a,b and table 1, the Langmuir model fitted well to illuminate the adsorptive experimental process on account of the great regression correlation of $R^2 > 0.99$. And the maximum removal capacity of Cr (VI) on nZVI@EG was 21.14, 24.04 and 26.67 mg g$^{-1}$ at different temperatures according to the Langmuir equation. On the contrary, low $R^2$ coefficients reflected that the experimental data in the adsorption process was unsuitable for Freundlich model. The results manifested that the removal process on Cr (VI) was proceeded as a monolayer adsorption, instead of a random multilayer adsorption on nZVI@EG. Furthermore, adsorptive kinetic experiments of Cr (VI) on nZVI@EG were managed to explore the impact on reaction time, using pseudo-first-order (equation (3.6)) and pseudo-second-order (equation (3.7)) kinetic models (figure 4c,d)

$$\frac{dq}{dt} = k_1(q_e - q_t) \tag{3.7}$$

and

$$\ln (q_e - q_t) = \ln (q_e) - k_2 t, \tag{3.8}$$

where $k_1$ and $k_2$ relate to the rate constants of pseudo-first-order and pseudo-second-order, respectively, and $q_e$ and $q_t$ (mg g$^{-1}$) represent the adsorptive amount of Cr (VI) on nZVI@EG at the equilibrium and random time.

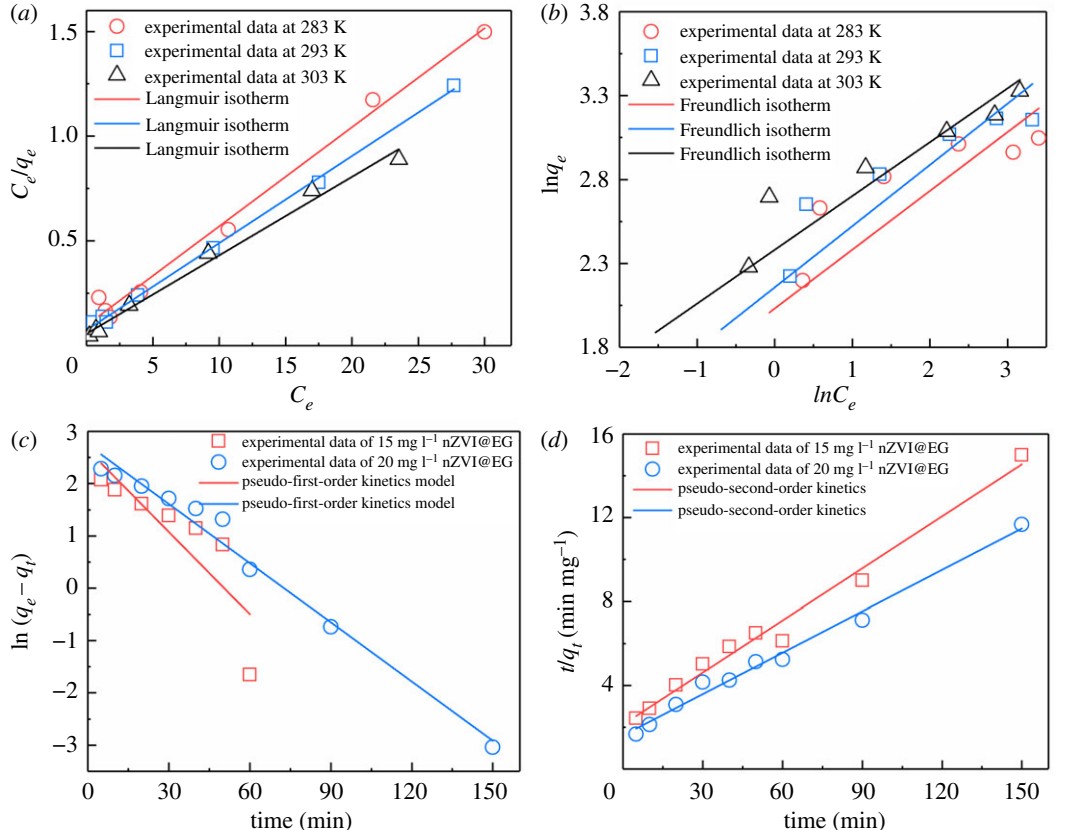

**Figure 4.** Theoretical model calculation. (a,b) Models of the Langmuir and Freundlich isotherms on the process of Cr (VI) adsorption of nZVI@EG at different reaction temperature. (c,d) Pseudo-first-order and pseudo-second-order adsorption kinetics models for Cr (VI) adsorption on nZVI@EG with different absorbent concentrations.

**Table 1.** Langmuir and Freundlich isotherm model constants and regression correlation coefficients.

| temperature (K) | Langmuir | | | Freundlich | | |
|---|---|---|---|---|---|---|
| | $Q_m$ (mg g$^{-1}$) | $K_L$ (l mg$^{-1}$) | $R^2$ | $K_F$ | $n_F$ | $R^2$ |
| 283 | 21.14 | 219.94 | 0.990 | 7.24 | 2.85 | 0.692 |
| 293 | 24.04 | 307.76 | 0.998 | 8.22 | 2.74 | 0.843 |
| 303 | 26.67 | 415.56 | 0.991 | 10.29 | 3.11 | 0.900 |

Based on computational kinetic parameters in table 2, the $R^2$ coefficients of pseudo-first-order model sustained at a low level with various initial concentrations, indicating that the experimental data were unable to match with this model. Nevertheless, the $R^2$ coefficients exceeded 0.99 based on pseudo-second-order kinetic model, implying that adsorptive experiments were well described, with equilibrium capacity of 21.18 and 24.29 mg g$^{-1}$, identified with the aforementioned calculated results. And ambient temperature was closely connected with the determination on reaction rate. Hence, the typical thermodynamic parameters were obtained to estimate the influence on temperature through the equations below [42]

$$\ln K_D = -\frac{\Delta H^0}{RT} + \frac{\Delta S^0}{R}, \tag{3.9}$$

$$K_D = \frac{C_0 - C_t}{C_t} \tag{3.10}$$

and

$$\Delta G^0 = \Delta H^0 - T\Delta S^0, \tag{3.11}$$

where $K_D$ is the adsorption equilibrium constant, $T$ (K) represents the reaction temperature, and $R$ is the ideal gas constant. $\Delta G^0$, $\Delta H^0$ and $\Delta S^0$ are Gibbs free energy, enthalpy and entropy, respectively.

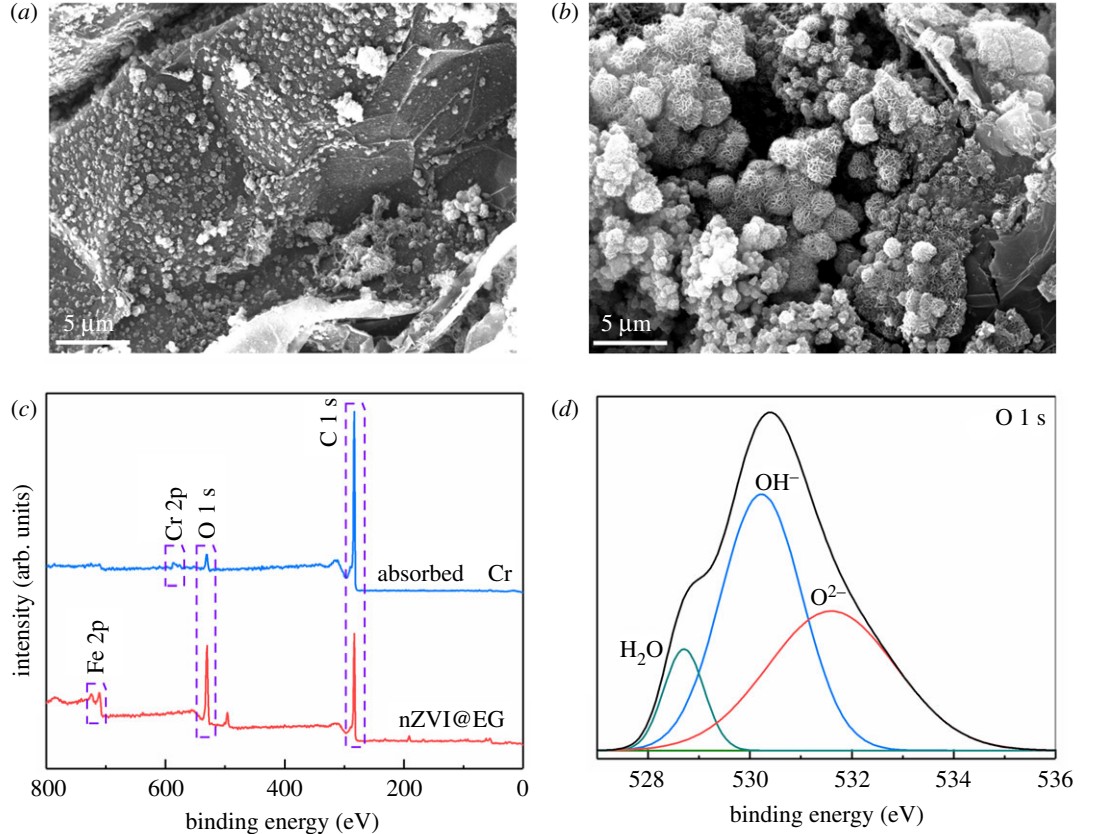

**Figure 5.** Mechanism analysis on the adsorption process. (*a,b*) SEM images on the surfaces of nZVI@EG after adsorption. (*c*) Wide-scan XPS spectrum of nZVI@EG before and after adsorption. (*d*) Typical O 1 s XPS spectrum of nZVI@EG after adsorption.

**Table 2.** Pseudo-first-order and pseudo-second-order kinetic parameters and regression correlation coefficients.

| $C_0$ (mg l$^{-1}$) | pseudo-first-order | | | pseudo-second-order | | |
|---|---|---|---|---|---|---|
| | $q_e$ (mg g$^{-1}$) | $k_1$ (min$^{-1}$) | $R^2$ | $q_e$ (mg g$^{-1}$) | $k_2$ (min$^{-1}$) | $R^2$ |
| 15 | 10.26 | 0.037 | 0.727 | 21.18 | 2.85 | 0.992 |
| 20 | 15.56 | 0.039 | 0.813 | 24.29 | 3.11 | 0.995 |

**Table 3.** Thermodynamic parameters.

| $\triangle G^0$ (KJ mol$^{-1}$) | | | | |
|---|---|---|---|---|
| 283 K | 293 K | 303 K | $\triangle H^0$ (KJ mol$^{-1}$) | $\triangle S^0$ (J mol K$^{-1}$) |
| −4.12 | −4.94 | −6.46 | 28.97 | 116.54 |

From the listed parameters in table 3, the values of $\Delta G^0$ at various ambient temperatures were negative, indicating the spontaneous adsorptive reaction. Moreover, the adsorption was an endothermic process, due to the positive values of $\Delta H^0$, with the requirement of extra energy input like heating up. And the increase on randomness was demonstrated by the positive value of $\Delta S^0$ for the interface physico-chemical reaction, presenting a tight affinity between Cr (VI) and nZVI@EG.

## 3.4. Adsorption mechanism

SEM images in figure 5*a,b* exhibit the surface topography of nZVI@EG after Cr (VI) adsorption. A mass of flocculent and spherical precipitates agglomerate formed on the surface, which were hydroxides generated in the reduction of hexavalent Cr to trivalent Cr and iron oxides/hydroxides. Furthermore,

the new peaks at the binding energy of around 576 and 587 eV were attributed to the formation of $Cr_2O_3$ and $Cr(OH)_3$ [43–45], manifesting that hexavalent chromium was attached to the appearance through redox and complexation reaction on nZVI@EG (figure 5c). The weak peak at nearly 580 eV occurred, implying the intense physical affinity of expanded graphite on Cr (VI). From the typical O 1 s XPS spectrum of figure 5g, the peaks located at 530, 531 and 533 eV were related to absorbed $H_2O$, $OH^-$ and $O^{2-}$, respectively, indicating the transformation of zero-valent ion to $Fe_2O_3$ or FeOOH, and Cr (VI) to $Cr(OH)_3$ or ferrochrome oxide. The reaction equations could be presented as follows:

$$Fe^0 + Cr_2O_7^{2-} + 14H^+ \rightarrow 2Cr^{3+} + 7H_2O + 2Fe^{3+} \quad (pH < 7), \tag{3.12}$$

$$Fe^0 + 2CrO_4^{2-} + 2H_2O \rightarrow 2Cr(OH)_3 \downarrow + 3Fe(OH)_3 \downarrow + 4OH^- \quad (pH \geq 7), \tag{3.13}$$

$$2Fe^0 + O_2 + 2H_2O \rightarrow 2Fe^{2+} + 4OH^-, \tag{3.14}$$

$$Fe^0 + 2H_2O \rightarrow Fe^{2+} + H_2 \uparrow + 2OH^-, \tag{3.15}$$

$$Cr_2O_7^{2-} + 6Fe^{2+} + 14H^+ \rightarrow 2Cr^{3+} + 6Fe^{3+} + 7H_2O \tag{3.16}$$

and

$$nCr^{3+} + (1-n)Fe^{3+} + 2H_2O \rightarrow Cr_nFe_{1-n}OOH \downarrow + 3H^+. \tag{3.17}$$

# 4. Conclusion

Summarily, nanoscale zero-valent iron was assembled effectually onto the surface of expanded graphite (nZVI@EG) through a simple way of liquid-phase reduction in our study. The findings of structure and morphology characterization indicated that the deposition of nZVI nanoparticles was firm with chemical binding, enlarging the specific surface areas and pore size distribution of the as-prepared absorbent. The equilibrium data on Cr (VI) adsorption of nZVI@EG matched highly with the Langmuir isotherms, following pseudo-second-order kinetics model in the meantime. And the whole adsorptive process was an endothermic reaction, and spontaneous, due to calculated thermodynamic parameters. Owing to the simultaneous effect on the physical adsorption of porous structure and chemical precipitation of zero-valent iron nanoparticles with strong reducing property, Cr (VI) could be converted into precipitates and adsorbed on the surface, with a great enhancement on adsorption capacity of nZVI@EG and removal efficiency of Cr (VI) in heavy-metal polluted wastewater.

Data accessibility. All of the data in this investigation have been reported in the paper and are freely available. Raw data, editable figures and tables input files are available in the Dryad Digital Repository at: doi:10.5061/dryad.rn8pk0pb2 [46].

Authors' contributions. Y.Q. and X.J. designed the experiments, guided the study and wrote the manuscript. X.C. and Y.Z. performed the preparation and adsorption experiments. X.C., Y.Q. and X.J. analysed the results and wrote the manuscript. All authors gave final approval for publication.

Competing interests. We declare we have no competing interests.
Funding. The financial support for this work provided by the Fundamental Research Funds for the Central Universities (grant no. WUT:2020IVA022) is gratefully acknowledged.

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
