## [Peer Review File · Royal Society Open Science]

Review History

RSOS-210801.R0 (Original submission)

Review form: Reviewer 1

Is the manuscript scientifically sound in its present form?

Yes

Are the interpretations and conclusions justified by the results?

Yes

Is the language acceptable?

Yes

Do you have any ethical concerns with this paper?

No

Have you any concerns about statistical analyses in this paper?

No

Recommendation?

Major revision is needed (please make suggestions in comments)

Comments to the Author(s)

In this manuscript, the authors synthesized nZVI@EG and characterized the composites in detail. The prepared composites were applied for the sorption and reduction of Cr(VI) to Cr(III) and the results showed that the composites could efficiently reduce Cr(VI) to Cr(III). After reading the manuscript, I think it can be accepted for publication after revision.

Special comments:

1. In the main text, I suggest the authors to revise "chromium (VI)" to Cr(VI).
2. In the Introduction section, two critical reviews should be added in the revised form such as: *The Innovation*, 2021, 2(1), 100076; *Biochar*. 2020, 2, 47-64.
3. It is better to add the effect of pH and ionic strength on Cr(VI) removal as this is helpful to understand the interaction mechanism from batch sorption results.
4. How about the stability of the composites in the removal of Cr(VI)?
5. The interaction mechanism should be discussed more detail in the revised form. Several important papers are helpful to improve the discussion of the results such as: *Environmental Research*, 2021, 196, 110349; *Chemosphere*, 2021, 262, 127901; *Environmental Science & Technology*. 2019, 53, 6454-6461.
6. How about the reusability of the composites in the removal of Cr(VI) from solutions?

Review form: Reviewer 2

Is the manuscript scientifically sound in its present form?

Yes

Are the interpretations and conclusions justified by the results?

Yes

Is the language acceptable?

Yes

Do you have any ethical concerns with this paper?

No

Have you any concerns about statistical analyses in this paper?

No

Recommendation?

Accept with minor revision (please list in comments)

Comments to the Author(s)

The authors have described a simply method to prepare a new type of absorbent materials, combining nanoscale zero-valent iron nanoparticles with expanded graphite with an enhancement on the performance properties to degrade different pollutants in aqueous solution. Characterization results and systematic condition experiments were presented to demonstrate the great removal effect, revealing the redox process as the adsorption mechanism with detailed simulation calculation. Thus, I would like to recommend the publication of the manuscript after a minor revision.

1 For the expanded graphite, the expanded volume is a significant measurable indicator. After heating, the acid treated graphite can be expanded to generate a vermiform and fluffy form. And what is the expanded volume? This should be added into the manuscript.

- 2 In the portion on the preparation of nZVI@EG, PEG and MF dispersant were added into the system. What is the purpose of these reagents?
- 3 The axis values and titles of the inserting figures in Figure 3a should be bold to achieve a clear expression.
- 4 What is the specific chemical reaction on the redox process between nZIV@EG and Cr(VI)? And Chemical equations may reflect the whole process obviously.

Decision letter (RSOS-210801.R0)

Dear Dr Jiao:

Title: Nanoscale Zero-valent Iron Loaded Vermiform Expanded Graphite for the Removal of Chromium (VI) from Aqueous Solution
Manuscript ID: RSOS-210801

The editor assigned to your manuscript has now received comments from reviewers. We would like you to revise your paper in accordance with the referee and Subject Editor suggestions which can be found below (not including confidential reports to the Editor). Please note this decision does not guarantee eventual acceptance.

Please submit your revised paper before 14-Jul-2021. Please note that the revision deadline will expire at 00.00am on this date. If we do not hear from you within this time then it will be assumed that the paper has been withdrawn. In exceptional circumstances, extensions may be possible if agreed with the Editorial Office in advance. We do not allow multiple rounds of revision so we urge you to make every effort to fully address all of the comments at this stage. If deemed necessary by the Editors, your manuscript will be sent back to one or more of the original reviewers for assessment. If the original reviewers are not available we may invite new reviewers.

RSC Associate Editor:
Comments to the Author:
(There are no comments.)

RSC Subject Editor:
Comments to the Author:
(There are no comments.)

Reviewers' Comments to Author:
Reviewer: 1

Comments to the Author(s)

In this manuscript, the authors synthesized nZVI@EG and characterized the composites in detail. The prepared composites were applied for the sorption and reduction of Cr(VI) to Cr(III) and the results showed that the composited could efficiently reduce Cr(VI) to Cr(III). After reading the manuscript, I think it can be accepted for publication after revision.

Special comments:

1. In the main text, I suggest the authors to revise "chromium (VI)" to Cr(VI).
2. In the Introduction section, two critical reviews should be added in the revised form such as: The Innovation, 2021, 2(1), 100076; Biochar. 2020, 2, 47-64.
3. It is better to add the effect of pH and ionic strength on Cr(VI) removal as this is helpful to understand the interaction mechanism from batch sorption results.
4. How about the stability of the composites in the removal of Cr(VI)?
5. The interaction mechanism should be discussed more detail in the revised form. Several important papers are helpful to improve the discussion of the results such as: Environmental Research, 2021, 196, 110349; Chemosphere, 2021, 262, 127901; Environmental Science & Technology. 2019, 53, 6454-6461.
6. How about the reusability of the composites in the removal of Cr(VI) from solutions?

Reviewer: 2

Comments to the Author(s)

The authors have described a simply method to prepare a new type of absorbent materials, combining nanoscale zero-valent iron nanoparticles with expanded graphite with an enhancement on the performance properties to degrade different pollutants in aqueous solution. Characterization results and systematic condition experiments were presented to demonstrate the

great removal effect, revealing the redox process as the adsorption mechanism with detailed simulation calculation. Thus, I would like to recommend the publication of the manuscript after a minor revision.

1 For the expanded graphite, the expanded volume is a significant measurable indicator. After heating, the acid treated graphite can be expanded to generate a vermiform and fluffy form. And what is the expanded volume? This should be added into the manuscript.

2 In the portion on the preparation of nZVI@EG, PEG and MF dispersant were added into the system. What is the purpose of these reagents?

3 The axis values and titles of the inserting figures in Figure 3a should be bold to achieve a clear expression.

4 What is the specific chemical reaction on the redox process between nZIV@EG and Cr(VI)? And Chemical equations may reflect the whole process obviously.

Author's Response to Decision Letter for (RSOS-210801.R0)

See Appendix A.

Decision letter (RSOS-210801.R1)

Dear Dr Jiao:

Title: Nanoscale Zero-valent Iron Loaded Vermiform Expanded Graphite for the Removal of Cr (VI) from Aqueous Solution

Manuscript ID: RSOS-210801.R1

It is a pleasure to accept your manuscript in its current form for publication in Royal Society Open Science. The chemistry content of Royal Society Open Science is published in collaboration with the Royal Society of Chemistry.

RSC Associate Editor
Comments to the Author:
(There are no comments.)

Reviewer(s)' Comments to Author:

Appendix A

Itemized Responses to Reviewer's Comments

Dear Editor and Reviewers:

Thank you for your email and the assessment of our submission.

We are grateful to the Reviewers for the excellent feedback and comments that have significantly improved the quality of the manuscript.

These comments are valuable and very helpful for revising and improving the manuscript, while helping to emphasize the significance of our research. We have studied the comments carefully and have made the appropriate corrections per recommendations. Revised portions of the manuscript are marked in red in the revised version. Responses to the Reviewers' comments and corresponding corrections are listed in a point-by-point reply as follows.

Please let us know should any other information and action be required from our side.

Responses to Reviewer 1

1. *“In the main text, I suggest the authors to revise “chromium (VI)” to Cr(VI)”*

R1. All the expressions of “chromium (VI)” have been revised to “Cr (VI)” in the main text.

2. *“In the Introduction section, two critical reviews should be added in the revised form such as: The Innovation, 2021, 2(1), 100076; Biochar. 2020, 2, 47-64.”*

R2. As the reviewer mentioned, the two critical reviews on typical absorbent materials for pollutants removal from aqueous solutions are meaningful to demonstrate the significance of heavy-metal ion removal and adsorption method, which have been added into the Introduction section as No.14 and 17 of references.

3. *“It is better to add the effect of pH and ionic strength on Cr(VI) removal as this is helpful to understand the interaction mechanism from batch sorption results.”*

R3. The effect of pH on the removal efficiency of Cr (VI) in aqueous solution has been presented in Figure 3a, showing great adsorption effect at the condition of low pH. In order to investigate the ionic strength on Cr (VI) removal, 0.1 mol/L of NaCl, Na₂SO₄, NH₄Cl and Na₂CO₃ have been added into the solution, respectively, with the result showing as below. From Figure R1, the removal efficiency at the presence of NH₄Cl achieved a high level over 90%, promoted by the acidic hydrolysis of ammonium ions. Inversely, the removal efficiency at the presence of Na₂CO₃ was low, due to the alkaline hydrolysis of carbonates, in accord with the pH testing experiments. Moreover, strong electrolyte ions such as Na⁺, Cl⁻ and SO₄²⁻, showed weak effect on the adsorption process.

Figure R1 The effect of various ions on the removal efficiency of Cr (VI) in aqueous solution.

4. *“How about the stability of the composites in the removal of Cr(VI).”*

R4. Before and after the adsorption of Cr (VI) in aqueous solutions, the mass change has been recorded to reflect the mechanical stability. And the results presented obvious weight increment, indicating few nanoparticles shedding and massive formation of sediments. In addition, the experiments were carried out at different pH, indicating good chemical stability in various aqueous environments. To verify the mechanical stability deeply, nZVI@EG were added into deionized water with powerful stirring. And the SEM images of the treated composite were presented in Figure R2, showing complete structure with abundant nanoparticles on the surface and no visible damage. Furthermore, the SEM images of the composite after different adsorption time were shown in Figure 5a and b, exhibiting great adsorption effect on Cr (VI) with the formation of numerous floccules.

Figure R2 SEM images of nZVI@EG after strong stirring in deionized water.

5. *The interaction mechanism should be discussed more detail in the revised form. Several important papers are helpful to improve the discussion of the results such as: Environmental Research, 2021, 196, 110349; Chemosphere, 2021, 262, 127901; Environmental Science & Technology. 2019, 53, 6454-6461.*

R5. The interaction mechanism has been discussed in detail, and the literatures were helpful to describe the reaction process. And the session of adsorption mechanism has been revised as below.

SEM images in Figure. 5a and b exhibited the surface topography of nZVI@EG after Cr (VI) adsorption. A mass of flocculent and spherical precipitates agglomerate formed on the surface, which were hydroxides generated in the reduction of hexavalent Cr to trivalent Cr and iron oxides/hydroxides. Furthermore, the new peaks at the binding energy of around 576 and 587 eV were attributed to the formation of Cr₂O₃ and Cr(OH)₃ [1-3], manifesting that hexavalent chromium was attached to the appearance through redox and complexation reaction on nZVI@EG (see Figure. 5c). The weak peak at nearly 580 eV occurred, implying the intense physical affinity of expanded graphite on Cr (VI). From the typical O 1s XPS spectrum of Figure 5g, the peaks located at 530, 531 and 533 eV were related to absorbed H₂O, OH⁻ and O²⁻, respectively, indicating the transformation of zero-valent iron to Fe₂O₃ or FeOOH, and Cr (VI) to Cr(OH)₃ or ferrochrome oxide. The reaction equations could be presented as follows.

1 Wang HH, Guo H, Zhang N, Chen ZS, Hu BW, Wang XK. 2019 Enhanced photoreduction of U(VI) on C₃N₄ by Cr(VI) and Bisphenol A: ESR, XPS, and EXAFS investigation. *Environ. Sci. Technol.* 53, 6454-5461. (doi:10.1021/acs.est.8b06913)

2 Zhu YL, He XY, Xu JL, Fu Zheng, Wu SY, Ni J, Hu BW. 2021 Insight into efficient removal of Cr(VI) by magnetite immobilized with *Lysinibacillus* sp. *JLT12: Mechanism and performance. Chemosphere* 262, 127901. (doi: 10.1016/j.chemosphere.2020.127901)

3 Qiu MQ, Liu ZX, Wang SQ, Baowei Hu. 2021 The photocatalytic reduction of U(VI) into U(IV) by ZIF-8/g-C₃N₄ composites at visible light. *Environ. Res.* 196, 110349. (doi:10.1016/j.envres.2020.110349)

6. *How about the reusability of the composites in the removal of Cr(VI) from solutions?*

R6. In order to study the reusability of nZVI@EG composite, cyclic tests have been conducted on the removal of Cr (VI), the result showing as following. In the cyclic test, after 2 g/L nZVI@EG adding into the solution for 90 min, nZVI@EG was taken out and immersed into 0.1 mol/L HCl solution to remove sediment for regeneration. And the whole process was regarded as one cycle. From Figure R3, after continuous adsorption tests for 8 cycles, the removal efficiency for Cr (VI) still maintained a high level over 80%, displaying great reusable performance.

Figure R3 Cyclic test for nZVI@EG on the removal efficiency of Cr (VI) in aqueous solution.

Responses to Reviewer 2

1. *“For the expanded graphite, the expanded volume is a significant measurable indicator. After heating, the acid treated graphite can be expanded to generate a vermiform and fluffy form. And what is the expanded volume? This should be added into the manuscript.”*

R1. The expanded volume of EG was 280 mL/g at the temperature of 800 °C, the average of three measurements.

2. *“In the portion on the preparation of nZVI@EG, PEG and MF dispersant were added into the system. What is the purpose of these reagents.”*

R2. Due to the natural hydrophobic property, expanded graphite (EG) was hard to be dispersed in the solvent. Thus, the addition of dispersants was beneficial to the dispersion of EG. Meanwhile, PEG and MF dispersant were conducive to the uniform deposition of nZVI nanoparticles on the surface.

3. *“The axis values and titles of the inserting figures in Figure 3a should be bold to achieve a clear expression.”*

R3. Figure 3a has been revised with bolder frame and fonts in the inserting figure as below.

Figure R4 The effects of environmental pH on the removal efficiency of Cr (VI) in aqueous solution, inserting the form distribution of Cr (VI) at different pH.

4. “What is the specific chemical reaction on the redox process between nZIV@EG and Cr(VI)? And Chemical equations may reflect the whole process obviously.”

R4. The whole adsorption process can be described that the Cr (VI) ions contacted the surface and multi-grade pores of the expanded graphite composite under the driving force of diffusion with physical adsorption. And then Fe⁰ nanoparticles reduced Cr (VI) to Cr (III), with the reaction equation as follows.